# Finding defects in glasses through machine learning

Simone Ciarella [1,7] ✉, Dmytro Khomenko [2,3,7] ✉, Ludovic Berthier [4,5], Felix C. Mocanu[1], David R. Reichman [2], Camille Scalliet [6] & Francesco Zamponi[1]

Structural defects control the kinetic, thermodynamic and mechanical properties of glasses. For instance, rare quantum tunneling two-level systems (TLS) govern the physics of glasses at very low temperature. Due to their extremely low density, it is very hard to directly identify them in computer simulations. We introduce a machine learning approach to efficiently explore the potential energy landscape of glass models and identify desired classes of defects. We focus in particular on TLS and we design an algorithm that is able to rapidly predict the quantum splitting between any two amorphous configurations produced by classical simulations. This in turn allows us to shift the computational effort towards the collection and identification of a larger number of TLS, rather than the useless characterization of non-tunneling defects which are much more abundant. Finally, we interpret our machine learning model to understand how TLS are identified and characterized, thus giving direct physical insight into their microscopic nature.

When a glass-forming liquid is cooled rapidly, its viscosity increases dramatically and it eventually transforms into an amorphous solid, called a glass, whose physical properties are profoundly different from those of ordered crystalline solids[1]. At even lower temperature, around 1 K, the specific heat of a disordered solid is much larger than that of its crystalline counterpart as it scales linearly rather than cubically with temperature. Similarly, the temperature evolution of the thermal conductivity in glasses is quadratic, rather than cubic[2–11]. A theoretical framework rationalizing such anomalous behavior was provided by Anderson, Halperin, and Varma[12] and by Phillips[13,14]. They argued that the energy landscape of amorphous solids contains many nearly-degenerate minima, connected by the localized motion of a few atoms, that can act as tunneling defects, called two-level systems (TLS)[15–19]. Since then, localized structural defects have been understood to play a crucial role in many other glass properties[20]. Understanding the microscopic origin of such localized defects and how to control their

density and physical properties is a major goal not only to improve our fundamental understanding of amorphous solids, but also for technological applications, such as optimizing the performance of certain quantum devices[21,22].

The development of particle-swap computer algorithms[23,24] has allowed the creation of computer glasses at unprecedentedly low temperatures. Combined with potential energy landscape exploration algorithms[25–34], this provides a powerful method to investigate the nature of defects in materials prepared under conditions comparable to experimental studies[35,36]. These tools have enabled direct numerical observation of TLS[35,37], confirming the experimental result[7,8,10,11,38] that the density of tunneling defects is strongly depleted as the kinetic stability of a glass increases. Similar results have been obtained for a different kind of defect, namely soft vibrational modes[39]. The direct detection of TLS revealed some of their microscopic features, namely that fewer atoms participate in the TLS of stable glasses[35]. It was also

[1]Laboratoire de Physique de l'École Normale Supérieure, ENS, Université PSL, CNRS, Sorbonne Université, Université de Paris, 75005 Paris, France. [2]Department of Chemistry, Columbia University, 3000 Broadway, New York, NY 10027, USA. [3]Dipartimento di Fisica, Sapienza Università di Roma, P.le A. Moro 2, I-00185 Rome, Italy. [4]Yusuf Hamied Department of Chemistry, University of Cambridge, Lensfield Road, Cambridge CB2 1EW, United Kingdom. [5]Laboratoire Charles Coulomb (L2C), Université de Montpellier, CNRS, 34095 Montpellier, France. [6]DAMTP, Centre for Mathematical Sciences, University of Cambridge, Wilberforce Road, Cambridge CB3 0WA, United Kingdom. [7]These authors contributed equally: Simone Ciarella and Dmytro Khomenko. ✉e-mail: simone.ciarella@ens.fr; dmytro.khomenko@uniroma1.it

shown that TLS is not in a one-to-one correspondence with soft harmonic[35,36,40] or localized[36,41] modes.

The main limitation of the direct landscape exploration method is its large computational cost, making it hard to construct the large library of defects needed for a robust statistical analysis of their physical properties. After accumulating a large number of inherent structures (IS), one must run a computationally expensive algorithm to find the minimum energy path connecting pairs of IS in order to determine if the pair forms a proper defect (e.g., to form a TLS, a defect must have quantum energy splitting within thermal energy at 1 K). The very large number of IS pairs detected makes it is impossible to characterize all of them. In previous works, some ad hoc filtering rules were introduced in order to identify candidate TLS and focus computational effort on them[35–37] but the success rate of such filters is poor. A considerable computational effort, which consisted in sampling ~$10^8$ IS, resulted in the direct detection of about 60 TLS. It is then obvious that most of the computational effort has been wasted in the study of pairs that form defects that do not tunnel at low temperatures. Looking for TLS is akin to finding the proverbial needle in a haystack.

In this paper, we demonstrate the relevance of machine learning techniques to predict with enhanced accuracy whether a pair of inherent structures forms a defect of a certain type. Recently, machine learning (ML) was shown to be extremely effective in using structural indicators to predict structural, dynamical, or mechanical properties of glassy solids[20,42–50]. Here, we use supervised learning to streamline the identification of defects. We focus in particular on the classical energy barrier and the quantum splitting associated with defects, which are relevant to identify TLS. Our study has two goals: (i) develop a faster way to identify TLS compared to the standard approach described below[35,37] in order to collect a statistically significant number of tunneling defects; (ii) determine the structural and dynamical features characterizing TLS as well as their evolution with glass preparation. To address (i) we show that our machine learning model can be trained in a few minutes using a small quantity of data, after which the model is able to identify candidate TLS with high speed and accuracy. To address (ii) we determine which static features are the most important for the model prediction. We show that TLS are not necessarily pairs of IS explored consecutively in the dynamics. We conclude by explaining how the ML model distinguishes TLS from non-TLS and how it is able to identify glasses prepared at different temperatures. While here we mostly focus on TLS, which is the rarest defect in glasses, our method should easily apply to other problems, such as supercooled liquid dynamics, plasticity, or devitrification of glassy solids.

## Results

In the following, we focus on the concreteness of the problem of detecting rare tunneling TLS. We also demonstrate that our method can successfully predict the classical energy barrier between two energy minima, with applications to the efficient detection of other kinds of defects.

### Machine-learning approach

The standard procedure[32,33,35] to identify TLS is sketched in Fig. 1a. The following steps aim at identifying potential candidates for TLS (see the Methods section for details): (1) Equilibrate the system at the preparation temperature $T_f$. Glasses with lower $T_f$ have increased glass stability. (2) Run molecular dynamics to sample configurations along a dynamical trajectory at the exploration temperature $T < T_f$. (3) Perform energy minimization from the sampled configurations to produce a time series of energy minima, or inherent structures (IS). (4) Analyze the number of transitions recorded between pairs of IS in the dynamical exploration of step 2, and select the pairs of IS that are explored consecutively.

Step 4 was necessary because it is computationally impossible to analyze all pairs of IS, as the number of pairs scales quadratically with the number of minima. The filter defined in step 4 was physically motivated by the fact that TLS tend to originate from IS that are not too distant in order to have a reasonable tunneling probability. As such it is likely that those pairs of IS get explored one after the other during the exploration dynamics in step 2. Overall, given $N_{IS}$ inherent structures, this procedure selects for $\mathcal{O}(N_{IS})$ pairs to be analyzed.

Once potential candidates are selected, the procedure continues as follows: (5) For each selected pair of IS, look for the minimum energy path and the classical barrier between them by running a minimum energy path-finding algorithm, such as the nudge elastic band (NEB) algorithm[51–53]. This provides the value of the potential energy $V(\xi)$ along the minimum energy path between the pair, where $0 \le \xi \le 1$ is the reaction coordinate. (6) Select pairs whose energy profile $V(\xi)$ has the form of a double well (DW), i.e. exclude paths with multiple wells. (7) Solve the one-dimensional Schrödinger equation:

$$-\frac{\hbar^2}{2md^2\epsilon}\partial_\xi^2\Psi(\xi) + V(\xi)\Psi(\xi) = \mathcal{E}\Psi(\xi), \tag{1}$$

where $\xi$ is a normalized distance along the reaction path $\xi = x/d$ and energy is normalized by a Lennard-Jones energy scale $\epsilon$, the effective mass $m$ and the distance $d$ are calculated as in ref. 35. We obtain the quantum splitting (QS) $E_{qs} = \mathcal{E}_2 - \mathcal{E}_1$ from the first two energy levels $\mathcal{E}_1$ and $\mathcal{E}_2$. The quantum splitting is the most relevant parameter for TLS

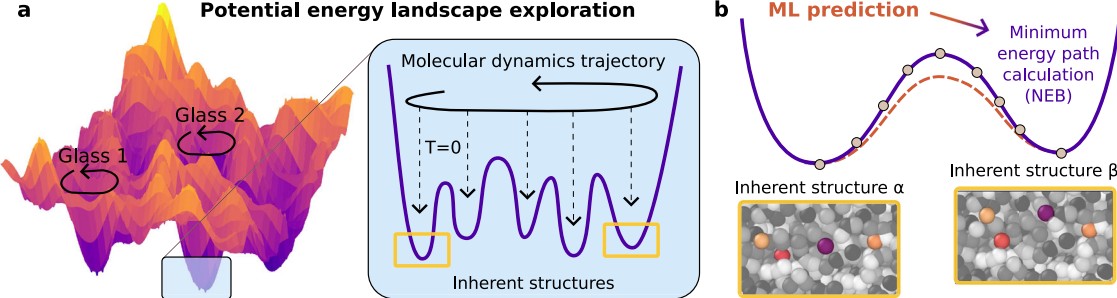

**Fig. 1 | Numerical search for two-level systems. a** Exploring the potential energy landscape: different glass samples define different metabasins in the rough landscape. (Inset) Each glass metabasin is explored via molecular dynamics simulations (black arrow) during which frequent energy minimization (dashed arrows) generates a large number of inherent structures (IS). Previous works restricted the search for candidate defects only to pairs of IS explored consecutively in the

dynamics. **b** Our machine-learning approach instead considers all IS pairs, irrespective of the dynamical exploration, and rapidly provides a robust prediction for their properties, such as quantum splitting. The candidates selected by the ML model are then analyzed via a minimum energy path-finding protocol (NEB algorithm) and their properties are computed exactly and compared with the ML prediction.

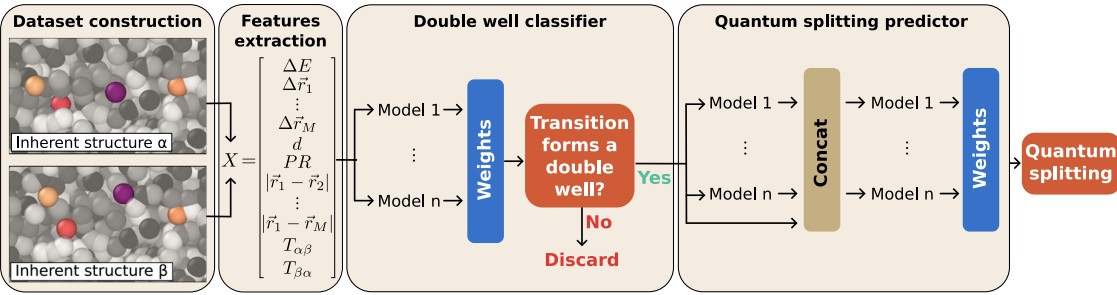

**Fig. 2 | Flowchart of the machine learning model.** The dataset is constructed by comparing all the pairs of inherent structures (IS), focusing on the $M$ atoms that displace the most between two IS (highlighted by colors: bright, resp. dark, indicate small, resp. large particle radius). Specific features are extracted to construct the input vector $X$. We then train a classifier to predict whether a pair of inherent structures forms a double well potential (DW) or not. The DW is finally processed using a multi-layer stacking strategy to predict the quantum splitting of the double well potential. Our pipeline analyses a given pair of IS in ~$10^{-4}$s.

because when $E_{qs} \sim T$ the system can transition from one state-to-the other via quantum tunneling[13]. In particular, since we choose to report the data in units that correspond to Argon[35], a double well excitation will be an active TLS at $T=1$ K when $E_{qs} < 0.0015\epsilon$, where $\epsilon$ sets the energy scale of the pair interactions in the simulated model.

Overall, since at low temperature the landscape exploration dynamics is slow, one would like to spend most of the computational time on steps 2–3 to construct a large library of pairs of IS. A first problem is that when the library of IS grows larger it takes a lot of time to perform steps 5–7. Moreover, the main bottleneck lies in the fact that most of the pairs that go through the full procedure turn out not to be TLS. The large computational time dedicated to steps 5–7 is thus wasted. Furthermore, many pairs of IS can be close but not sampled consecutively during the dynamics, owing to the high-dimensional nature of the potential energy landscape.

We now introduce our machine learning (ML) approach to the problem, whose main advantage is to consider all pairs of IS as TLS candidates. As shown below, our approach can detect TLS which are otherwise excluded in step 4. We distinguish two phases: training and deployment. Our supervised training approach, detailed in the Methods section and sketched in Fig. 2, takes just a few hours of training on a single CPU. It requires an initial dataset of $\mathcal{O}(10^4)$ full NEB calculations, whose collection is the most time-consuming part of the training phase. Once training is complete, the ML model can be deployed to identify new TLS.

Its workflow is similar to the standard one, with some major improvements. It proceeds with the following steps: (1)–(3) The first 3 steps are similar to the standard procedure to obtain a collection of inherent structures from a dynamical exploration. (4) Apply the ML model to all possible pairs of IS to predict which pairs form a DW potential. (5) Apply the ML model to predict the quantum splitting (QS) for all predicted DW and filter out the configurations that are not predicted to be TLS by the ML model. (6)–(8) For the pairs predicted to be TLS by the ML model only, run the NEB algorithm and select the pairs that form a DW potential. Solve the one-dimensional Schrödinger equation in order to obtain the exact value of the quantum splitting.

In the Methods section, we provide details on how steps 1–3 are performed: glass preparation, exploration of the potential energy landscape via molecular dynamics simulations and minimization procedure, as well as NEB computation. We also explain how it is possible to use steps 4-5 as a single shot or as an iterative training approach, see Methods Sec. IV H.

Importantly, the well-trained ML model has two significant advantages over the standard approach. First, $\mathcal{O}(N_{IS}^2)$ pairs of IS are scanned to identify TLS, compared to a much smaller number $\mathcal{O}(N_{IS})$ in the standard procedure. Second, if a pair of IS passes step 5 and goes through the full procedure it is very likely to be a real TLS. As a consequence, by using the ML approach one can spend more time doing steps 2–3 to produce new IS, since fewer pairs pass step 5. At the same time, for any given number of IS, the ML approach can analyze all possible pairs and is, therefore, able to identify many more TLS, as we demonstrate below.

## Quality of the machine learning prediction

In refs. 35,36, the authors analyze a library of 14,202, 23,535, and 117,370 pairs of inherent structures for a continuously polydisperse system of soft repulsive particles, equilibrated at reduced temperatures $T_f = 0.062$, $0.07$, and $0.092$, respectively. The standard approach described in Sec. II A leads to the identification of 61, 291, and 1008 TLS for the three temperatures, respectively. Note that this approach uses pairs of IS that are selected by the dynamical information contained in the transition matrix between pairs of IS[35]. This was done to filter out all non-DW potentials. For all pairs in this small subset, the quantum splitting was then calculated.

Instead, the ML approach starts by independently evaluating the relevant information contained in each IS and constructs all possible combinations, even for pairs that are not dynamically connected in the landscape exploration. Following the steps discussed in Sec. II A the model is then able to predict which of all pairs form a DW, as well as the value of their quantum splitting, very accurately. From a quantitative perspective, this means that the same dynamical trajectories now contain many more TLS candidates in the ML approach compared to the standard approach.

In this section we briefly describe the flowchart of the model summarized in Fig. 2. A detailed description of the machine learning model is provided in the Methods section. First, for all the available IS, we evaluate a set of static quantities that we use to construct the input features for each pair of IS. By convention, we label the IS with $\alpha = 1, \dots, N_{IS}$ by increasing potential energy $E_1 < \dots < E_{N_{IS}}$ where $E_\alpha$ is the potential energy of IS$_\alpha$. We use the convention that $\alpha < \beta$. The input feature for a pair $\alpha\beta$ of IS consists in the potential energy difference $\Delta E = E_\beta - E_\alpha$ between the two minima, the displacements $\Delta \vec{r}_i$ of the $M$ particle which displace the most between the two IS (labeled with $i = 1, \dots, M$ by decreasing displacement), as well as the relative positions between those $M$ particles, the total displacement of particles $d$, and participation ratio $PR$, all computed by comparing the two IS. We also use as input, the number of transitions recorded in the dynamical exploration $T_{\alpha\beta}$ (from the lowest energy IS to the highest, in our convention) and $T_{\beta\alpha}$ (from highest to lowest IS). See Methods Sec. IV E for more details. We then apply it in series two model ensembles. Model ensembling consists in averaging the output of different ML models to achieve better predictions compared to each of them separately. The first ensemble is trained to classify DW (Methods, Sec. IV F), which is a necessary condition for TLS. A DW is defined when the minimum energy path between the two IS resembles a quartic potential, as sketched in Fig. 1b. For the pairs that are predicted to be DW, a second model ensemble (Methods, Sec. IV G) is used to predict the quantum splitting (QS), which determines if the pair is a TLS or not.

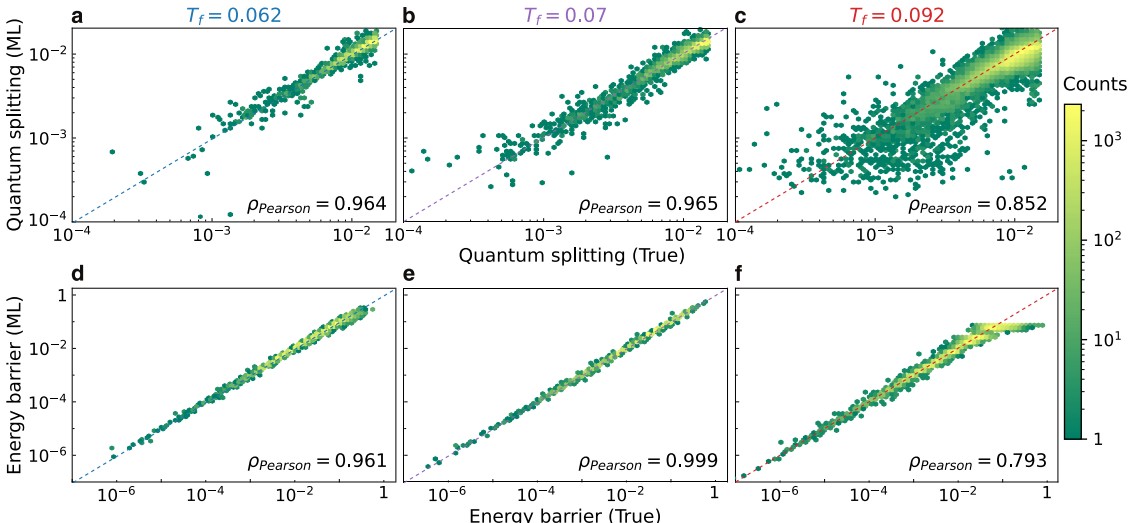

**Fig. 3 | Machine-learning prediction for the quantum splitting and classical energy barrier between pairs of inherent structures (IS). a–c** Quantum splitting and **d–f** energy barrier predicted by the ML model compared to the exact value. The model was not trained on these IS pairs. Glass stability decreases from left to right: glasses are equilibrated at (**a, d**) $T_f = 0.062$, (**b, e**) $T_f = 0.07$, and (**c, f**) $T_f = 0.092$. The ML model was trained on 7000, 10,000, and 30,000 samples respectively, using the information on the $M = 3$ particles with the largest displacements. All models were trained for ~10 hours of single CPU time.

To showcase the performance of the ML model we report in Fig. 3a–c the exact QS calculated from the NEB procedure, compared with the value predicted by the model. We see that the data concentrates around the diagonal, indicating good correlation between true and predicted values. The Pearson correlation reported in the figure provides a quantitative measure for the correlation. The train/test split is performed by randomly selecting 10% of the pairs to be used only for the evaluation. We have trained three independent models to work at the three different glass preparation temperatures. As explained in the Methods (Sec. IV E), the model needs the information about the $M \ll N$ particles that displace the most between the two IS only to achieve the excellent precision demonstrated in Fig. 3. In the Supplementary Fig. 2 we show that the optimal value is $M = 3$, i.e., information on only three particles is needed for the model to identify TLS, confirming the low participation ratio in TLS. Furthermore, the models have been trained using the smallest number of samples, randomly selected from all the IS pairs available, that allow the model to reach its top performance. We have also performed an analysis of the optimal training time. Details on these points are provided in Supplementary Note 1. The performances presented in Fig. 3 are achieved by training the model for ~10 hours of single CPU time, but we also show in Supplementary Note 1 that it is possible to already achieve >90% of this performance by training the ensemble for only 10 minutes.

The ML approach introduced here is also easily generalizable to target other quantities related to state-to-state transition, such as excitations and higher energy effects. We modified the quantum splitting predictor to instead predict the classical energy barrier between two IS states. If the minimum energy path between two IS forms a DW, we define the classical energy barrier as the energy difference between the saddle point and the lowest energy minimum. In Fig. 3d–f, we report the value of the energy barrier predicted by the ML model compared to the exact value calculated from the NEB procedure. We use the same hyperparameters and features as the quantum splitting predictor. Such a high performance demonstrates that our ML approach can predict other types of transitions between states, associated with distinct kinds of defects.

## Capturing elusive TLS with machine learning
We now use ML in order to speed up the TLS search. This highly efficient method allows us to collect a library of TLS of unprecedented size, generated from numerical simulations with the same interaction potential as in ref. 35, see Methods for its definition. First of all, we reprocess the data produced to obtain the results presented in ref. 35 with our new ML method based on iterative training (Methods, Sec. IV H), obtaining new information about the connection between TLS and dynamics. Next, we perform ML-guided exploration to collect as many TLS as possible. This sizable library of TLS allows us to perform for the first time a detailed statistical analysis of TLS and compare their distribution to the distribution of double wells. We perform this analysis for glasses of three different stabilities. Finally, we discuss the microscopic features of TLS not only by looking at their statistics, but also by analyzing what the ML model has learned, and how it expresses its predictions.

Prior to this paper, it was not possible to evaluate all the IS pairs collected in ref. 35. For this reason, the authors introduced a filtering rule based on the assumption that high transition rates during the dynamic landscape exploration are a good indicator that the minimum transition path between two IS forms a double well. Therefore, ref. 35 discarded all pairs $\alpha\beta$ of IS such that the number of jumps $T_{\alpha\beta}$ (from low to high energy IS) and $T_{\beta\alpha}$ (high to low) during the MD exploration is smaller than four, i.e., $\min(T_{\alpha\beta}, T_{\beta\alpha}) < 4$. This reduced the number of pairs to 14,202, 21,109, and 117,339 for glasses prepared at $T_f = 0.062$, 0.07, and 0.092, respectively. In order to have comparable data at the three temperatures, for $T_f = 0.092$ we only consider a subset of glasses corresponding to 30920 IS pairs.

The results of the TLS search are summarized in the red columns of Tab. 1. Overall, the standard procedure reaches a rate of TLS found per NEB calculation of $4 \times 10^{-3}$, $13 \times 10^{-3}$, and $8 \times 10^{-3}$ with increasing $T_f$. We compare these numbers with those obtained with our iterative training procedure applied to the same data, see green columns of Tab. 1. We immediately notice two major improvements. First, the overall number of TLS that we find from the same dataset is more than twice larger. Second, the ratio of TLS per NEB calculation is more than 15 times larger, corresponding to $62 \times 10^{-3}$, $211 \times 10^{-3}$, and $194 \times 10^{-3}$ with increasing $T_f$.

We conclude that the iterative ML approach is much more efficient than the standard procedure, and also that TLS does not necessarily have a large dynamical transition rate, since the dynamical-filtering approach discards more than half of them.

## Differences between DW and TLS

With our ML-driven exploration of the energy landscape we can focus the numerical effort on DW and favorable TLS candidates, while processing a larger number and/or longer exploration trajectories. This allows us to consider a larger set of independent glasses of the same type as those treated in ref. 35, which is particularly relevant for ultrastable glasses generated at the lowest temperature $T_f = 0.062$. While in ref. 35 the collection of 61 TLS required more than 14,000 NEB calculations, we are able to identify 872 TLS running 11 iterations of iterative training using only a total of 5500 NEB calculations in addition to the ~6000 used for pretraining. In the next section we analyze these results to discuss the nature of TLS.

The database of glasses that we analyze with iterative training contains 5 times more IS than in ref. 35, and we find up to 15 times more TLS by running around half of the NEB calculations. We report in Fig. 4 the results from this extended TLS search. In Fig. 4a, we report predicted and true values for the quantum splitting, with a background color coding for the confusion matrix. The threshold is set by the fact that TLS have $E_{qs} < 0.0015\epsilon$. The number of data points in each quadrant is reported in the inset. The horizontal dashed lines highlight the percentage of true TLS detected by running the NEB algorithm for all points with a predicted quantum splitting below the line. Due to false negatives, it is better to also consider transitions slightly above the TLS threshold. We see that all TLS are identified by considering only the pairs that are predicted to be within twice the quantum splitting

threshold of TLS. All TLS thus are safely detected by running 2484 NEB calculations, out of 4147 DW in total. In Fig. 4b, we report the cumulative density of TLS quantum splitting $n(E_{qs})$, which according to the TLS model scales as $n(E_{qs}) \sim n_0 E_{qs}$ at low $E_{qs}$[12,14]. We show indeed that $n(E_{qs})/E_{qs}$ converges to a plateau $n_0$ for small $E_{qs}$. We have recorded $n_0 = 0.67$, 4.47 and 25.14 in units of $\epsilon^{-1}\sigma^{-3}$. This is approximately $10^{23}$, $10^{24}$ and $10^{25}$ eV$^{-1}$ cm$^{-3}$ in Ar units. In refs. 35,37, we discuss the comparison between numerical and experimental TLS densities. The ML approach allows us to collect significantly better statistics compared to ref. 35, confirming that the TLS density $n_0$ decreases by several orders of magnitude from hyperquenched to ultrastable glasses. Lastly, in Fig. 4c, we report the histograms of the number of TLS and DW per glass at the three temperatures. We see that when the glasses are ultrastable ($T_f = 0.062$) most of the glasses have very few TLS. Conversely, poorly annealed ($T_f = 0.092$) glasses show a very unbalanced distribution, with a few glasses that contain most of the DW and TLS.

## Interpretation of the ML model

The ML model contains precious information about the distinctive structure of TLS. First, the present and previous works[7,8,10,11,35,37,38] find that the density of TLS decreases upon increasing glass stability, which in our simulations is controlled by the preparation temperature $T_f$. Thus, one may also expect temperature-dependent TLS features. In the Supplementary Note 4 we show that when the ML model is trained on glasses prepared at $T_{train}$ and deployed on glasses prepared at $T_{prediction} \neq T_{train}$ there is only a minor performance drop and the model is able to perform reasonably well. This implies that the model captures distinctive signatures of TLS that do not depend strongly on the preparation temperature. Yet, we also show in the Supplementary Note 4 that it is very easy to train another ML model to predict the temperature itself and eventually add it to the pipeline.

Overall, the ML model is not only able to capture the different microscopic features of TLS, but it can also suggest what the specific influence of each feature is. To interpret this information we calculate Shapley values[54] for each input feature and report them in Fig. 5 for the quantum splitting predictor (a) and the double well classifier (b). The features are ranked from the most important (top) to the less important (bottom). We first discuss the quantum splitting predictor Fig. 5a. The horizontal axis reports their impact (SHAP value) on the model output: large positive SHAP values predict on average a high value of the quantum splitting (QS). The data points are colored following the value of the feature itself. The most important feature is the classical energy splitting $\Delta E$ corresponding to the potential energy difference between the two IS. In our model, a large value of classical splitting $\Delta E$ (red) implies a large QS, i.e., non-TLS transitions. The second most

### Table 1 | Comparison of the standard procedure with our iterative training approach

|       | Standard procedure (ref. 35) | | Iterative training (this paper) | |
|-------|------------|-----------|------------|------------|
| $T_f$ | # TLS | # NEBs | # TLS | # NEBs* |
| 0.062 (Ar) | 61 | 14,202 | 156 | 2500 |
| 0.062 (NiP) | 28 | 14,202 | 59 | 2000 |
| 0.07 (Ar) | 291 | 21,109 | 1057 | 5000 |
| 0.07 (NiP) | 46 | 21,109 | 152 | 4000 |
| 0.092 (Ar) | 245 | 30,920 | 776 | 4000 |
| 0.092 (NiP) | 28 | 30,920 | 123 | 6000 |

Analysis of data collected in ref. 35. We report results for different glass stabilities, decreasing from top to bottom, using Argon units (Ar)[30] and NiP metallic glass parameters (NiP)[27]. The standard procedure finds less than half of the TLS found with the ML and is computationally much more expensive.

*This number does not include the data that we use to pre-train the model.

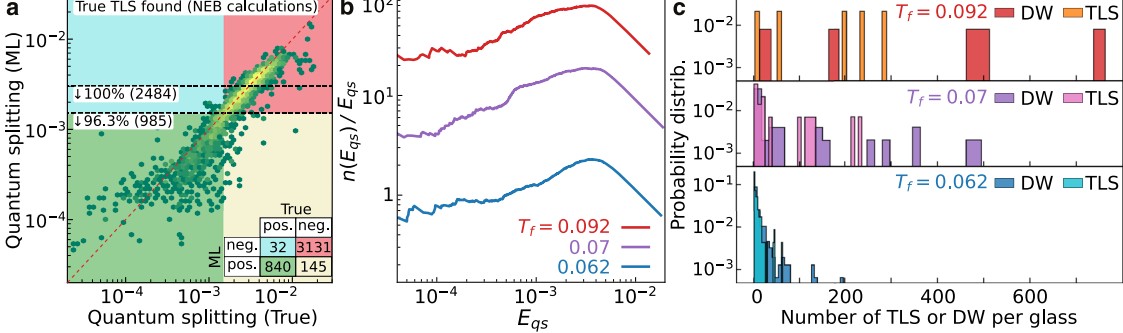

**Fig. 4 | The machine-learning-driven exploration identifies an unprecedentedly large number of two-level systems. a** We compare the model predictions to the calculated values at the end of our iterative training, for stable glasses $T_f = 0.062$. The confusion matrix is color-coded in the background and reported on the bottom right of **a**. Horizontal dashed lines report the percentage of TLS that is predicted below that value of quantum splitting, showing that >95% of the TLS are

within twice the TLS threshold of $0.0015\epsilon$. **b** Cumulative distribution of quantum splitting $n(E_{qs})$ divided by $E_{qs}$. **c** Histograms of the number of TLS and DW per glass, at the three preparation temperatures $T_f = 0.092$, 0.07, and 0.062 from top to bottom. We have considered 5, 30, and 237 glasses, respectively. Results are reported for Ar units.

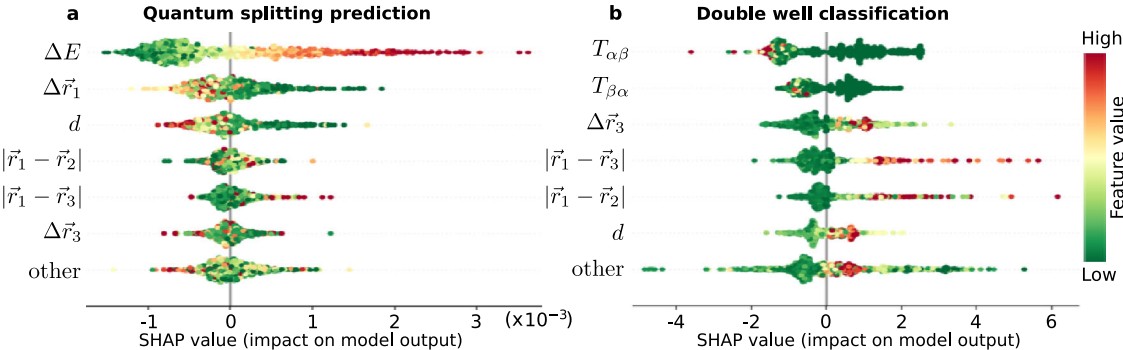

**Fig. 5 | Determining important features to predict quantum splittings and classify double well potentials.** Importance of the different features for the ML models **a** predicting the quantum splitting (QS), and **b** classifying double wells, both at $T_f = 0.062$. The features include the single particle displacements $\Delta \vec{r}_i$ (which decrease with increasing label $i$), their relative position compared to the most displacing particle $|\vec{r}_1 - \vec{r}_i|$ and the number of recorded transitions from the lowest to highest energy IS in the dynamic exploration $T_{\alpha\beta}$ (and $T_{\beta\alpha}$ from high to low-energy IS). The features are ordered from top to bottom by decreasing importance. Each point corresponds to a single IS pair, with a color coding for the feature value (red: high, green: low). The points are spread vertically for readability. The feature impact on the model output is given on the horizontal axis: large SHAP values predict large QS values, low SHAP values predict low QS (more likely to be a TLS).

important feature is the largest single particle displacement $\Delta \vec{r}_1$, which has to be larger than a threshold corresponding to $0.3\sigma$ in order to predict a TLS (low SHAP, hence low QS). The total displacement $d$ is the third most important and shows a similar effect. All the remaining features have a less clear and much smaller effect on the model prediction and they only collaborate collectively to the final QS prediction. Details on features definition are provided in the Methods. In the Supplementary Note 2 we show that it is possible to obtain very good performance even when removing some of the features with the largest Shapley values, which means that the ML interpretation is not unique.

According to this Shapley analysis we explain the ML prediction in the following way: the energy difference $\Delta E$ between two IS is the main predictor for the quantum splitting, and it has to be small for TLS. Then, the largest particle displacement $\Delta \vec{r}_1$ is necessary to understand if the two IS are similar and what is their stability (we show in the Supplementary Note 2 that $\Delta \vec{r}_1$ is the most important feature to identify the glass stability). Then the total displacement $d$ complements this information and gives local information about the displacements of the other particles. Lastly, all the other inputs provide fine tuning to refine the final prediction and are discussed in more detail in the Supplementary Note 2. Interestingly, in the Supplementary Note 2 we also show that even without the two most important features, the ML approach can still identify TLS candidates reasonably well.

## Microscopic features of TLS
We have shown that by following a ML-driven approach it is possible to collect a significant library of TLS for any preparation temperature. However it may be useful to discuss alternative strategies to rapidly identify TLS. In general, since TLS are extremely rare defects[7,8,10,11,38] a filtering rule is necessary in order to reduce the number of possible candidates. In particular, ref. 35–37 proposed to use the number of transitions recorded between two IS during the MD exploration, and to exclude pairs that are not explored consecutively. This is based on the assumption that DW (and consequently TLS) correspond to IS pairs that are close to each other and should thus be visited consecutively in the dynamics (non-zero number of recorded transitions).

Instead, we prove in Fig. 6 that a filter based on dynamical information only is a poor predictor. In Fig. 6a, we report the distribution of the number dynamical transitions recorded between two inherent structures $\alpha$ and $\beta$, $T_{\alpha\beta} + T_{\beta\alpha}$ (both from low to high and high to low-energy IS). We report three curves measured for TLS, DW, and all pairs,

measured at $T_f = 0.062$. While the slowly decaying tail of TLS and DW suggests that they often exhibit a large transition rate, actually most TLS and DW are formed by IS with no recorded transitions between them in the dynamic exploration.

Our interpretation is that even though the transition from one IS to the other is favorable, the landscape has such a large dimensionality ($3N$) that even very favorable transitions may never take place in a finite exploration time. This issue can become more severe when the trajectories are shorter, for example if the exploration is performed in parallel. We confirm this observation with the results reported in Tab. 1, where we have used our iterative training approach to re-analyze the data of ref. 35, including pairs with no recorded transition, and found many more TLS.

We conclude that even though the number of recorded transitions is the most important feature to predict which pair forms a double well, as seen in Fig. 5b, a filter based solely on them still misses many pairs of interest and therefore is not the most efficient.

In Fig. 6b, we focus on the distribution of the classical splitting $\Delta E$, or energy difference between the two IS. When $\Delta E$ is large, the transition path between IS rarely forms a DW, or a TLS (red region). On the other hand, there are many pairs with a very small $\Delta E$ which are not necessarily more likely to be DW or TLS, hence the yellow region (could be any of TLS, DW, or non-DW). Ultimately we find a 'sweet spot' (green region), where TLS are more frequent. The ML model also captures this feature, as seen from the SHAP parameter of $\Delta E$ in Fig. 5a. The next most important feature according to the ML model is the largest particle displacement $\Delta \vec{r}_1$, reported in Fig. 6c. When it is larger than $-0.8\sigma$ we rarely find TLS and DW, but we do not find them also when $\Delta \vec{r}_0 < 0.3\sigma$. The second row in Fig. 5a confirms that the ML model has discovered this feature. In Fig. 6d, we report the total displacement $d$. If $d > 0.9\sigma$ the pair is so different that it is not likely to be a TLS or DW, while this probability increases for smaller $d$. In Fig. 6e, we report the distribution of off-diagonal elements $\Delta_0$, measured using the WKB approximation as explained in ref. 35. We find that the distribution obtained from TLS and DW scales as $1/\Delta_0$, in good agreement with the standard TLS model[12].

Finally, if one is interested in identifying TLS in a 'quick and dirty' way, we propose to use the number of recorded transitions to filter DW from non-DW, and then to select a sweet spot for the classical energy splitting and the displacements for selecting optimal TLS candidates.

## Discussion
In this paper we have introduced a machine-learning approach to explore complex energy landscapes, with the goal of efficiently

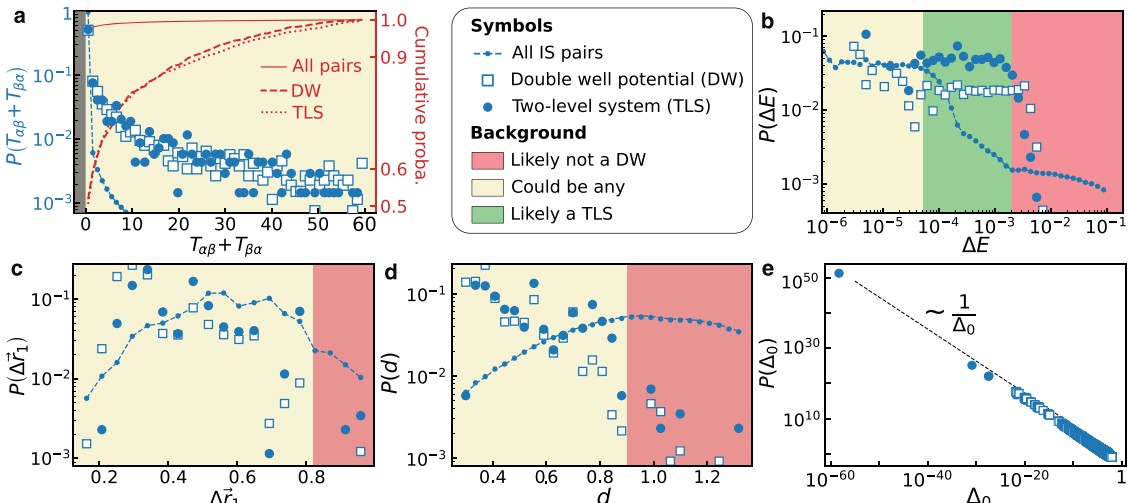

**Fig. 6 | Microscopic features of two-level systems and double-well potentials in ultrastable glasses.** Probability distributions of **a** number of recorded transitions between the two inherent structures $T_{\alpha\beta} + T_{\beta\alpha}$, **b** classical energy splitting $\Delta E$, **c** largest particle displacement $\Delta \vec{r}_1$, **d** total displacement $d$ and **e** WKB off-diagonal element $\Delta_0$. We color-coded in red the regions of parameters where we do not expect to find TLS, which instead concentrate in the green regions. The points in the yellow regions could be any of TLS, DW, or non-DW. The green regions could serve as an alternative way to rapidly identify TLS. Data for stable glasses created at $T_f = 0.062$.

locating double wells (DW) and two-level systems (TLS). We demonstrate that it is possible to use ML to rapidly estimate the quantum splitting of a pair of inherent structures (IS) and accurately predict if a DW is a TLS or not. We also show that our ML approach can be used to predict very accurately the energy barrier between pairs of IS, which would be useful to analyze supercooled liquid dynamics, or for the response to a mechanical deformation. Overall, this approach allows us to collect a library of defects of unprecedented size that would be impossible to obtain without the support of ML.

The ML model uses as input the information calculated from a pair of inherent structures. After just a few minutes of supervised training it is able to infer with high accuracy the quantum splitting of any new pair of inherent structures. We establish that our ML model based on model ensembling and gradient boosting is fast and precise. Its efficiency allows us to introduce an iterative training procedure, where we perform a small batch of predictions and then retrain the model.

After performing statistical analysis over the unprecedented number of TLS collected with our method, we have discovered that many DW and TLS are not consecutively visited during the dynamical exploration. We reanalyzed the data collected for the study of ref. 35 and found that more than half of the TLS were missed, because the corresponding IS were not visited consecutively and the pair was consequently discarded. Our ML approach not only finds more than twice the number of TLS from the same data, but it also requires significantly fewer calculations. We conclude that ML significantly improves the existing approaches. The ML method allows us to propose a 'quick and dirty' way to predict TLS: a) use $T_{\alpha\beta}$, $T_{\beta\alpha}$ for predicting DW; b) for those which are DW, use the classical energy splitting between the two IS to predict which are TLS.

We also discuss the microscopic nature of DW and TLS. We perform a Shapley analysis to dissect the ML model and understand what it learns, and we compare this with the extended statistics of TLS that we are able to collect. We find that the quantum splitting is mostly related to the classical energy splitting and the displacements of the particles. Overall, the Shapley analysis suggests that TLS are characterized by one particle that displaces between 0.3 and 0.9 of its size, while the total displacement and the energy difference between the two states remains small. The local structure around the particle is not as important, nor is the number of times we actually see this transition during the exploration dynamics.

Lastly we investigate the effect that glass stability (equivalent to the preparation temperature in our simulations) has on double wells and TLS. The ML model learns that at higher temperatures all the pairs are characterized by more collective rearrangements, but TLS are similar for any preparation temperature.

Ultimately, since our ML approach is extremely efficient in exploring the energy landscape and is easy to generalize to target any type of state-to-state transition (as we show for the energy barriers), we hope that our method will be used in the future to analyze not only TLS, but also many other examples of phenomena related to specific transitions between states in complex classical and quantum settings.

## Methods

### Glass-forming model

We study a three-dimensional polydisperse mixture of $N = 1500$ particles of equal mass $m$. The particle diameters $\sigma_i$ are drawn from the normalized distribution $P(0.73 < \sigma < 1.62) \propto 1/\sigma^3$. Two particles $i$ and $j$ separated by a distance $r_{ij}$ interact via the repulsive pair potential

$$v(r_{ij})/\epsilon = (\sigma_{ij}/r_{ij})^{12} + c_0 + c_2(r_{ij}/\sigma_{ij})^2 + c_4(r_{ij}/\sigma_{ij})^4, \quad (2)$$

only if $r_{ij} \leq 1.25\sigma_{ij}$, with the non-additive interaction $\sigma_{ij} = 0.5(\sigma_i + \sigma_j)$ $(1 - 0.2|\sigma_i - \sigma_j|)$. The polynomial coefficients $c_0, c_2, c_4$ ensure continuity of $v$ and its first two derivatives at the interaction cutoff. We study the system at number density $\rho = 1$ in a cubic box with periodic boundary conditions. We express energies and lengths in units of $\epsilon$ and the average diameter $\sigma$, respectively. Times measured during molecular dynamics (MD) simulations are expressed in units of $\sqrt{m\sigma^2/\epsilon}$. We make two choices for physical units following past work,[35] one corresponds to Ar atoms[55]: $m = 6 \times 10^{-26}$ kg, $\epsilon/k_B = 478$ K, $\sigma = 3.41 \times 10^{-10}$ m and $\tau = 1.08 \times 10^{-12}$ s; the other is for a NiP alloy[56]: $m = 1.02 \times 10^{-25}$ kg ($^{62}$Ni isotope), $\epsilon/k_B = 4447$ K, $\sigma = 2.21 \times 10^{-10}$ m and $\tau = 2.86 \times 10^{-13}$ s.

### Glass sample preparation

We fully equilibrate $n_g = 5, 50, 200$ independent configurations of the liquid at preparation temperatures $T_f = 0.092, 0.07, 0.062$, respectively. We do so employing the hybrid MD/particle-swap Monte Carlo algorithm described in ref. 24. The algorithm alternates between blocks of $5N$ attempts of particle-swap Monte Carlo moves and short

MD runs of duration $t_{MD} = 0.1$ to thermalize the liquid efficiently. Glassy samples are then created by rapidly cooling the equilibrium configurations to $T = 0.04$ using regular MD with a Berendsen thermostat[57]. The preparation temperature $T_f$ is thus Tool's fictive temperature[58] and characterizes the degree of stability of the glass: glasses prepared at a lower $T_f$ are more stable. We compare these $T_f$ with characteristic temperature scales. The mode-coupling crossover temperature is $T_d = 0.1$, and the extrapolated experimental glass transition temperature, where the relaxation time is 12 decades larger than at the onset of glassy dynamics, is $T_g = 0.067$[24]. The lower $T_f = 0.062$ glasses are ultrastable, while the higher $T_f = 0.092$ are hyperquenched.

### Energy landscape exploration and transition matrix $T_{\alpha\beta}$

We use classical molecular dynamics (MD) to explore the potential energy landscape of the glasses. We run MD simulations at $T = 0.04$ in the NVE ensemble with a time step $dt = 0.01$. Configurations along the MD trajectory are quenched to the closest potential energy minimum, or inherent structure (IS), every $\tau_{quench} = 0.2, 0.1, 0.05$ (for glasses prepared at $T_f = 0.062, 0.07, 0.092$, respectively) using the conjugate gradient method. The quench period $\tau_{quench}$ is chosen such that two consecutive quenches typically reach different IS, separated by one energy barrier. For each $n_g$ glass sample we perform 100, 100, 200 MD trajectories starting from different initial velocities, each lasting 40000, 100000, 10000 time steps (low to high $T_f$). For $T_f = 0.092$ we used a subset of the data obtained in[35].

The transition matrix elements $T_{\alpha\beta}$ count how many times $IS_\alpha$ and $IS_\beta$ are visited consecutively, i.e., $IS_\alpha$ is reached at time $t$, and $IS_\beta$ at time $t + \tau_{quench}$. Overall, $T_{\alpha\beta}$ is a number that counts the number of transitions observed from $IS_\alpha$ to $IS_\beta$, with no physical dimensions. Since this quantity depends on the specific trajectories collected, it varies for different quenching rates and simulation times. Here, we demonstrate that one advantage of our ML approach over a brute-force approach, as employed in ref. 35, is to consider all IS pairs, not only those visited consecutively in the MD trajectory (characterized by $T_{\alpha\beta} > 0$), as potential TLS. This expands massively the pool of candidates, while ensuring that computational effort is targeted to IS pairs that are likely to be TLS.

To analyze the transition between two IS we compute the multi-dimensional minimum energy path separating them. This is done by the nudged elastic band (NEB) method[51,52] implemented in the LAMMPS package. We use 40 images to interpolate the minimal energy path, that are connected by springs of constant $\kappa = 1.0\epsilon\sigma^{-2}$, and use the FIRE algorithm in the minimization procedure[51,53]. The NEB algorithm outputs a one-dimensional potential energy profile $V(\xi)$ defined for the reduced coordinate $\xi$, between the two minima.

### Quantum splitting computation

We extrapolate the potential obtained from the NEB, defined only between the two minima, to obtain a full double well potential. We used a linear extrapolation of the NEB reaction path. Let us denote $\mathbf{r}_1$ and $\mathbf{r}_2$ the coordinates of the particles in the first two images of the system along the reaction path ($\mathbf{r}_1$ is an energy minimum). We extrapolate the potential $V$ starting from $\mathbf{r}_1$ and measuring the potential energy of the configuration moving in the direction $\mathbf{r}_1 - \mathbf{r}_2$. We perform a similar extrapolation at the other minimum.

Once the classical potential $V(\xi)$ is obtained by extrapolation as discussed above, we solve numerically the Schrödinger Eq. (1) using a finite difference method. In general, the Laplacian term should take into account curvature effects along the reaction coordinates, as discussed in ref. 35. For simplicity, we neglect these effects and use the standard second derivative along the reaction coordinate.

### Dataset and features construction for ML approach

The first step of our machine learning approach is the evaluation of a set of static quantities for all the available IS. This set consists of:

potential energy, particle positions, averaged bond-orientational order parameters[59] determined via a Voronoi tessellation, that we denote $\{q_2, q_3...q_{12}\}$, and finally particle radii. The cost of this operation scales as the number of available states $N_{IS}$, but we use these quantities to calculate the features of $\sim N_{IS}^2$ pairs. A detailed analysis reported in Supplementary Note 1, shows that the bond-orientational parameters and the particle sizes are not very useful for the ML model. Since their calculation is slower than all the other features, we do not include them in the final version of the ML approach.

To construct the input features for each pair of IS we combine the information of the two states evaluating the following: (1) Energy splitting $\Delta E$: energy difference between the two IS. (2) Displacements $\Delta\vec{r}_i$: displacement vector of particle $i$ between the two configurations. When used in this context index $i$ increases with decreasing displacement $\Delta\vec{r}_1 > \Delta\vec{r}_2 > ....$ (3) Total displacement $d$: total distance between the two IS defined as $d^2 = \sum_i |\Delta\vec{r}_i|^2$. Participation ratio $PR$: defined as $PR = (d^2)^2 / (\sum_i |\Delta\vec{r}_i|^4)$. (4) Distance from the displacement center $|\vec{r}_1 - \vec{r}_i|$: we measure the average distance of particle $i$ from the center of displacement $\vec{r}_1$, identified as the position of the particle that moves the most. This quantity identifies the typical size of the region of particles that rearrange. (5) Transition matrix $T_{\alpha\beta}$ (resp. $T_{\beta\alpha}$): number of times the dynamics explores consecutively the lowest (resp. highest) then the highest (resp. lowest) energy minimum.

The crucial step of the feature construction is that we can reduce the number of features by considering only the $M$ particles whose displacement is the largest between pairs of IS. We make this assumption because we expect that the low temperature dynamics is characterized by localized rearrangements involving only a small fraction of the particles[7,8,10,11,35,38]. In Supplementary Note 1, we confirm this assumption by showing that the ML model achieves optimal performances even when $M$ is very small. So, the choice of $M \ll N$ makes the ML model computationally effective without any performance drop.

### Double well classifier

A necessary condition in order for a pair of IS to be a TLS is that the transition between the pair forms a double well (DW) potential. A DW is defined when the minimum energy path between the two IS resembles a quartic potential, as sketched in Fig. 1b. The final goal of the ML model is to predict the quantum splitting (QS) of the pair and identify pairs with low QS. The first obstacle in the identification of TLS is that DW represent only a small subgroup of all IS pairs. For instance in ref. 35, only ~0.5% of all the IS pairs are DW at the lowest temperature. It is then mandatory to filter out pairs that are not likely to be a DW.

In the machine learning field there are usually many different models that can be trained to achieve similar performances, with complexity ranging from polynomial regression to deep neural networks. Here, we perform model ensembling and use ensembles both for DW classification and QS prediction. Model ensembling consists in averaging the output of different ML models to achieve better predictions compared to each of them separately. In practice, we use the publicly available AutoGluon library[60]. In this approach, we train in a few minutes a single-stack ensemble that is able to classify DW with >95% accuracy. In the Supplementary Note 1, we justify this choice of ML model and provide details on performances and hyperparameters.

In particular, we get the best results using ensembles of gradient boosting models[61,62], which have proven to be the optimal choice in estimating barrier heights of chemical reactions computed with similar methods[63]. The gradient boosting approach predicts the probability $p(y_i) = G(x_i)$ that a pair $x_i$ is a DW, where $y_i = 1$ if the pair is a DW and 0 otherwise. It is based on a series of $m = 1, ..., n_{WL}$ weak learners $W_m$ that minimize the log-loss score

$$\frac{1}{n}\sum_i^n y_i \log[p(y_i)] - (1 - y_i) \log[1 - p(y_i)], \tag{3}$$

where $n$ is the number of IS pairs. In this approach, each of the weak learners $W_m$ attempts to improve over the result of its predecessor by predicting the residual $h_{m-1}(x_i) = y_i - W_{m-1}(x_i)$. The final prediction thus becomes

$$p(y_i) = \sum_{m=1}^{n_{\mathrm{WL}}} W_m(x_i | W_{l<m}). \tag{4}$$

This approach usually outperforms random forests[64], where the prediction is just the average over the weak learners $p(y_i) = 1/n_{\mathrm{WL}} \sum_{m=1}^{n_{\mathrm{WL}}} W_m(x_i)$. We then build a stack of gradient boosting models such that the final prediction is given by $p(y_i) = 1/n_{\mathrm{GB}} \sum_{k}^{n_{\mathrm{GB}}} c_k G_k(x_i)/\sum_k c_k$, which is the weighted average over the $n_{\mathrm{GB}}$ models in the ensemble with learnable weights $c_k$. Overall, our DW classifier turns out to be very accurate and rapid, achieving >95% accuracy after only 10 minutes of training. As such, it is convenient to use it to filter out the pairs that do not require the attention of the QS predictor because they cannot be TLS anyway.

### Quantum splitting predictor

We want to predict the quantum splitting of a pair of IS for which the features discussed in Sec. IV E have been computed. We need this prediction to be very precise, because we know that a pair can be considered a TLS when $E_{qs} < 0.0015\epsilon$, but $E_{qs}$ can vary significantly so errors may be large. In the Supplementary Note 1 we show that models such as deep neural networks and regression are not stable or powerful enough to achieve satisfying results. We thus follow the same strategy introduced for DW classification, by using model ensembling[60] and gradient boosting[61,62,64,65]. Compared to the DW predictor, each model $G_k$ in the ensemble now performs a regression task by predicting $E_{qs,k} = G_k(x_i)$. We then construct a multi-layer stack (schematized in Fig. 2), where the prediction of the first stack $E_{qs}^{(0)} = 1/n_{\mathrm{GB}} \sum_{k}^{n_{\mathrm{GB}}} c_k G_k(x_i)/\sum_k c_k$ is concatenated to $x_i$ and used as input for the following stack. At the same time, we also perform $k$-fold bagging[66], which consists in splitting the data in $k$ subsets used to train $k$ copies of each model with different data. This has shown to be particularly effective in improving the prediction for small datasets[60].

In order to train the model we first collect a set of $E_{qs}$ examples. The size of this training set is discussed in the Supplementary Fig. 2 where we find that the minimum number is around $10^4$. We can use some of the data already collected in previous work in ref. 35 for the training. Moreover, since we are interested in estimating with more precision the lowest values of $E_{qs}$ we train the model to minimize the following loss function

$$\mathcal{L} = \frac{\sum_{i=1}^{n} w_i \left( E_{qs,\text{true}} - E_{qs,\text{predicted}} \right)^2}{n \sum_{i=1}^{n} w_i}, \tag{5}$$

which is a weighted mean-squared error. The weights correspond to $w_i = 1/E_{qs,\text{true}}$ in order to give more importance to low $E_{qs}$ values. We thus train our model to provide a very accurate prediction of the value $E_{qs}$ for any given pair. Once the model is trained it takes only ~$10^{-4}$ s to predict the QS of a new pair (compared to 1 minute to run the standard procedure). If we predict a value $E_{qs} < 0.0015\epsilon$, then we have identified a TLS much faster.

### Iterative training procedure

We finally introduce an approach to optimally employ our ML model to process new data: the iterative training procedure. To produce the results reported in Fig. 3 we trained the model once using a subset of the already available data. This is a natural way to proceed when the goal is to process new data that are very similar to the training set, and the training set is itself large enough. However, since the goal of the proposed ML model is to ultimately drive the landscape exploration

**Table 2 | Computational time needed to perform our ML approach, on an Intel i9-9980HK CPU**

| | |
|---|---|
| Collection of an IS pair | 50 s |
| NEB+Schrödinger for a IS pair | $10^3$ s |
| Predict if an IS pair is a DW | $10^{-5}$–$10^{-4}$ s |
| Predict $E_{qs}$ for an IS pair | $10^{-5}$–$10^{-4}$ s |
| Train DW classifier | $10^3$–$10^4$ s |
| Train QS predictor | $10^3$–$10^4$ s |

and collect new samples, the single-training approach may encounter two types of problems. First, at the beginning there may be not enough data and, second, the findings of the model do not provide any additional feedback.

To solve both problems we introduce the iterative training procedure. The idea of iterative training is to use the predictive power of ML to create and expand its own training set, consequently enhancing its performance by iteratively retraining over the new data. Compared to standard active learning methods, iterative training does not focus on new samples with the highest model uncertainty, but instead it iterates the predictions on the samples below the threshold of interest. Details on the method and parameters are discussed in Supplementary Note 3. In practice, we start from a training set of $K_0 \sim 10^3$–$10^4$ randomly selected pairs to have an initial idea of the relation between input and output. We then use the ML approach outlined in Fig. 2 to predict the $K_i = 500$ pairs with the lowest QS. For these TLS candidates, we perform the full procedure to calculate the true QS and determine whether the pair is a DW or a TLS. In the Supplementary Fig. 6, we report the result of this procedure when we process a new set of trajectories from the same polydisperse soft sphere potential as in ref. 35. In general the first few iterations of iterative training have a poor performance. In fact, we find that >70% of the first $K_i$ pairs are actually non-DW. After collecting additional $K_i$ measurements, we retrain the model. We report in Tab. 2 the average time for each step of the ML procedure. The retraining can be done in ~10 min, after which the model is able to predict the next $K_i$ pairs with lowest QS. Overall, to process $N_{\mathrm{ISpairs}}$ we estimate that the computational time of the iterative approach is $t_i = \left[ K_0 \cdot 10^2 + N_{iter}(K_i \cdot 10^2 + 10^3 + N_{\mathrm{ISpairs}} \cdot 10^{-5}) \right] s$. If $N_{\mathrm{ISpairs}} > 10^9$ it is possible to significantly reduce $t_i$ by permanently discarding the worst pairs, but this is not needed here. We iterate this procedure $N_{iter}$ times, until the last batch of $K_i$ candidates contains less than 1% of the total number of TLS. We believe that continuing this iterative procedure would lead to the identification of even more TLS/DW, but this is out of the scope of this paper.

## Data availability

The processed data are available at zenodo.org/record/8026630.

## Code availability

The codes that support the findings of this study are available at github.com/SCiarella/TLS_ML_exploration.

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

## Acknowledgements

We thank E. Flenner, G. Folena, M. Ozawa, J. Sethna, S. Elliott, G. Ruocco, and W. Schirmacher for useful discussions. This work was granted access to the HPC resources of MesoPSL financed by the Region Ile de France and the project Equip@Meso (reference ANR-10-EQPX-29-01) of the program Investissements d'Avenir supervised by the Agence Nationale pour la Recherche. This project received funding from the European Research Council (ERC) under the European Union's Horizon 2020 research and innovation program, Grant no. 723955—GlassUniversality (FZ), and from the Simons Foundation (#454933, L.B., #454955, F.Z., #454951 D.R.) and by a Visiting Professorship from the Leverhulme Trust (VP1-2019-029, L.B.). C.S. acknowledges support from the Herchel Smith Fund and Sidney Sussex College, University of Cambridge.

## Author contributions

S.C. developed the machine learning model and approach; D.K., C.S. performed the simulations; D.K., L.B., C.S., D.R.R., F.Z. developed the simulation model; S.C., D.K., F.C.M., F.Z. analyzed the data; S.C., D.K., L.B., F.C.M., D.R.R., C.S., F.Z. wrote the paper; L.B., C.S., D.R.R., F.Z. provided resources and supervision.

## Competing interests

The authors declare no competing interests.
