## [Peer Review File · Nature Communications]

Finding defects in glasses through machine learningREVIEWER COMMENTS

Reviewer #2 (Remarks to the Author):

See attached pdf file for my report.

Reviewer #4 (Remarks to the Author):

The paper is overall well written and the results are convincing. The ML used is simple but this is not the point of this paper; rather it provides a clear use case where ML can unlock science in a justified way, by allowing to pre-filter specific pairs of configurations which are relevant to describe glass's dynamics.

I would overall recommend acceptance of this paper provided the more detailed points below can be addressed. That being said, there may remain a question of impact of this paper (more precisely: what will be the impact of gaining access to more TLS pairs) which I am not fully confident on.

More detailed comments:

1. Even though this is first and foremost a physics paper, I think that some details on the ML procedure and results could be explained a bit further, for instance (apologies if I missed some of those in the text):

- a. Giving some high level details of the type of models used in the main text rather than the name of the package used
- b. Reporting the accuracy/recall of the DW-classifier in the main text
- c. Discussing the train/test split used for training and evaluating the model
- e. It looks like the data is naturally described in log space -- is this how the models were trained? If not, can this be an issue in relation with numerical accuracy? In particular can you discuss the larger errors near the left parts of the diagrams in Fig 1?
- e. I was quite strongly surprised to see the MLP results in the Appendix, in particular the fact that the network does not match the mean of the prediction -- could you discuss this further?

2. On Figure 1

- a. Could you discuss why the errors are higher at high temperature? Does it connect to Fig 4.c?
- b. Could you add a quantitative measure of correlation on the plots?
- c. Are the errors larger at lower energies and is that a potential concern?

3. On Figure 2:

- a. Could you explain what data is the confusion matrix based on and maybe giving number for it (otherwise it is very hard to read); also it is my understanding that the confusion matrix can only be based on pairs where $T_{ij}+T_{ji} \leq 4$, can we make an argument that it extrapolates to the general case?
- b. I also did not understand what the topmost horizontal line was indicating.

3. On Figure 6.1,

- a. Would it make sense to report the cumulative fraction of split under a given $T_{ij}+T_{ji}$? That way one could check how to set the cutoff to gain a given fraction, yet not incur quadratic complexity.
- b. I suppose the reason why there are splits for larger $T_{ij}+T_{ji}$ is because of the MD-induced walk in the energy landscape backtracking, is that correct? It may be worth saying in words

(but maybe already said and I missed it).

4. On the "quick and dirty" method

- a. Is there really a usecase for it given how quickly one can train your model?
- b. If there is, is it possible to come-up with a fixed criterion and evaluate its performance?
- c. (related) Would it be possible on Table 2 to report the typical time taken by MD to generate a candidate IS pair? Unless it is \ll the NEB time, then the approach in this paper is less useful -- is this correct?

5. On the "iterative training", how is that different from what would generally fall under "active learning"?

Referee’s Report on 410056 “Finding defects in glasses through machine learning,” Simone Ciarella, Dmytro Khomenko, Ludovic Berthier, Felix C. Mocanu, David R. Reichman, Camille Scalliet, and Francesco Zamponi

There has been a long history of trying to find two level systems (TLS) using numerical simulations, but this has been difficult because TLS are so rare with a typical concentration of 100 ppm. In a previous paper (Ref. 35), the authors made a significant advance and were able to identify a statistically significant number of TLS using swap Monte Carlo. In the current manuscript, the authors have employed machine learning (ML) to efficiently explore the potential energy landscape of a 3D polydispersed mixture of soft spheres. In particular, they are able to use ML to look at all pairs of inherent structures (IS) in an effort to find double well (DW) potentials. Once DWs that are likely to be TLS are identified, they solve the 1D Schrodinger equation to calculate the quantum energy splitting. As a result they find considerably more TLS, roughly 2 to 5 times more TLS than their previous approach. This is a real tour-de-force and a significant advance. In my opinion, the paper should be published once the following questions and comments are addressed.

Comments and questions:

1. The authors find a flat distribution of quantum energy splittings at low energies which is consistent with the standard model of TLS. Since they are using the numbers for argon and NiP, can they state this distribution in terms of a density of states with physical units, e.g., number of TLS/(eV-cm³)? This would allow direct comparison with the typical values measured in glasses.
2. Since the authors find the energy barriers associated with their double wells, what is the distribution of energy barriers for their TLS? The standard model of TLS assumes a flat distribution of the WKB exponent $\lambda \sim \sqrt{2MV}d/\hbar$ where V is the barrier height, M is the mass of the tunneling entity, and d is the distance the tunneling entity moves. Or equivalently, what is the distribution of the tunneling matrix element $\Delta_o = \hbar\omega_o e^{-\lambda}$? It would be interesting to compare this to the standard TLS model assumption that $P(\Delta_o) \sim 1/\Delta_o$.
3. For their TLS, what is the distribution of classical splittings ΔE ? I am assuming that ΔE is the difference between the minima of a pair of IS. In the standard TLS model, the distribution of asymmetry energies ΔE , i.e., the difference between the minima of the double well, is flat.
4. The authors talk repeatedly about the dynamical transition T_{ij} between the i th IS and the j th IS, but they never really explain how this transition is calculated. Presumably they run their simulations and occasionally quench to find an IS. But how do they look at dynamical transitions between any given pair of IS? In their previous paper (Ref. 35), they talk about allowing their MD simulation to rapidly visit distinct IS, but there is no such discussion in this paper. Furthermore, since they investigate all pairs of IS, it is not at all clear that it is easy to get from one IS to its partner in the pair. The members of a pair of IS could be very different, making it hard to travel from one to the other. Some discussion needs to be added to explain how dynamical transitions are explored.
5. Following up on the previous point, what exactly is T_{ij} ? Is it a transition rate? In other words, what are the units of T_{ij} ?

Typos:

1. Page 1, top of second column: “has allowed to create computer glasses” would be better phrased “has allowed the creation of computer glasses”
2. Page 2, 1st column: “associated to defects” should be “associated with defects”
3. Page 4, 1st column: “each pairs of IS” should be “each pair of IS”
4. Page 5, 1st column, last paragraph: In this paragraph the authors talk about discarding IS pairs when the number of transitions is less than 4. Then they say that this is based on the assumption that high transition rates between two IS is a good indicator of a DW. This is a bit confusing because one sentence talks about low transition rates and the next sentence talks about high transition rates. So it would be better perhaps to start by stating the assumption that high transition rates between two IS is a good indicator of a DW, and then say that therefore, they discard those pairs which have a low transition rate.
5. Page 5, 2nd column, last paragraph: “focus the numerical effort to DW” should be “focus the numerical effort on DW”
6. Page 6, 2nd column, end of 1st paragraph: “with few glasses that contains” should read “with a few glasses contain”. Notice that there are 2 typos in this sentence.
7. Page 7, Figure 5(a): $|\vec{r}_0 - \vec{r}_1|$, $|\vec{r}_0 - \vec{r}_2|$, and $\Delta\vec{r}_2$ are not defined in the text or in the caption. These should be defined in the caption.
8. Page 7, Figure 5(b): T_{ij} and T_{ji} are not defined until later in the text. Therefore, they should be defined in the caption.
9. Page 7, 1st column: The classical energy splitting ΔE should be defined in the text where it is used. The definition is buried in the Methods section at the end of the paper.
10. Pages 7-8: I am reiterating an earlier comment, but at the bottom of page 7 and the top of page 8, the authors talk about transition rates between IS and that many pairs of IS have no transition. But the authors never explain how they look for transitions.
11. Page 8, 1st column, line 2: “When ΔE is large we rarely” should have a comma so that it reads, “When ΔE is large, we rarely”
12. Page 8, 2nd column: end of 2nd paragraph: “batch of prediction” should be “batch of predictions”
13. Page 9, 1st column, 1st line under METHODS: “polydisperse mixture” should be “polydispersed mixture”
14. Page 9, 2nd column: The Berendsen thermostat should have a reference to the appropriate paper or book.
15. Page 10, 1st column: What are q_2 and q_{12} ? These are never defined.
16. Page 10, 2nd column: Does QS stand for quantum splitting? The authors should define QS.

17. Page 12, Reference 12: “P. w. Anderson” should be “P. W. Anderson”

Once these comments, questions and typos have been addressed, I recommend that the paper be published in Nature Communications.

Reviewed by Clare Yu.

Reply to Reviewer #2

There has been a long history of trying to find two level systems (TLS) using numerical simulations, but this has been difficult because TLS are so rare with a typical concentration of 100 ppm. In a previous paper (Ref. 35), the authors made a significant advance and were able to identify a statistically significant number of TLS using swap Monte Carlo. In the current manuscript, the authors have employed machine learning (ML) to efficiently explore the potential energy landscape of a 3D polydispersed mixture of soft spheres. In particular, they are able to use ML to look at all pairs of inherent structures (IS) in an effort to find double well (DW) potentials. Once DWs that are likely to be TLS are identified, they solve the 1D Schroedinger equation to calculate the quantum energy splitting. As a result they find considerably more TLS, roughly 2 to 5 times more TLS than their previous approach. This is a real tour-de-force and a significant advance. In my opinion, the paper should be published once the following questions and comments are addressed.

We thank the Reviewer for their time, valuable comments and appreciation of our work. The points raised by the Reviewer helped us to improve the quality of our manuscript. Below, we reply point-by-point to all concerns raised. Most importantly, for each point we have revised the manuscript accordingly, with changes highlighted in red to facilitate review.

Comments and questions:

1. The authors find a flat distribution of quantum energy splittings at low energies which is consistent with the standard model of TLS. Since they are using the numbers for argon and NiP, can they state this distribution in terms of a density of states with physical units, e.g., number of TLS/(eV-cm³)? This would allow direct comparison with the typical values measured in glasses.

We thank the Reviewer for this suggestion. We have added the TLS density estimation in physical units for the data reported in Figure 4b: "We have recorded $n_0 = 0.67, 4.47$ and 25.14 in units of $\epsilon^{-1}\sigma^{-3}$. This is approximately $3.988 \times 10^{23}, 2.624 \times 10^{24}$ and 1.476×10^{25} eV⁻¹ cm⁻³ in Ar units. In Refs. [35,37], we discuss the discrepancy between numerical and experimental TLS densities."

Indeed, we discussed the dimensional analysis in more detail in Ref. 35 and 37, where we also speculated on possible reasons for the larger TLS density found in simulations compared to experiments. We found in Ref. 37 that part of the increased density of defects is related to polydispersity. The remaining difference could be related to a lack of many-body interactions in our simple models, or the difference between our classical exploration protocol and the quantum exploration of the landscape which occurs in experiment.

In this work, the focus was on improving the statistics of TLS, and we continue to observe numbers that are consistent both with a depletion of defects with increasing glass stability, and with quasi-universality. We believe that these are important milestones on the way towards a quantitative microscopic theory of these defects.

2. Since the authors find the energy barriers associated with their double wells, what is the distribution of energy barriers for their TLS? The standard model of TLS assumes a flat distribution of the WKB exponent $\lambda \sim \sqrt{2MVd}/\hbar$ where V is the barrier height, M is the mass of the tunneling entity, and d is the distance the tunneling entity moves. Or equivalently, what is the distribution of the tunneling matrix element $\Delta_0 = \hbar\omega_0 e^{-\lambda}$? It would be interesting to compare this to the standard TLS model assumption that $P(\Delta_0) \sim 1/\Delta_0$.

We now report in Fig. 6(e) the probability distribution of off-diagonal elements $P(\Delta_0)$, confirming the $\sim \frac{1}{\Delta_0}$ scaling for both DW and TLS (the latter are represented by blue circles). We added

a sentence to the main text: "In Fig. 6(e) we report the distribution of off-diagonal elements Δ_0 , measured using the WKB approximation as explained in Ref. [35]. We find that the distribution obtained from TLS and DW scales as $1/\Delta_0$, in good agreement with the standard TLS model [12]."

We had a plot of this in the SI of Ref. [35] which confirmed the $1/\Delta_0$ scaling, however we could not previously separate TLS from DW.

3. For their TLS, what is the distribution of classical splittings ΔE ? I am assuming that ΔE is the difference between the minima of a pair of IS. In the standard TLS model, the distribution of asymmetry energies ΔE , i.e., the difference between the minima of the double well, is flat.

First of all, we have clarified the definition of the classical energy splitting ΔE which is indeed "the potential energy difference between two IS". We report the distribution of classical splitting ΔE in Fig 6.b. In particular, the blue circles represent the distribution for TLS. We see that the distributions assume a constant value $P(\Delta E < 10^{-3}\epsilon) \sim 10^{-2}$, suggesting that the distribution of asymmetry energy is flat, as predicted by the standard TLS model.

4. The authors talk repeatedly about the dynamical transition T_{ij} between the i th IS and the j th IS, but they never really explain how this transition is calculated. Presumably they run their simulations and occasionally quench to find an IS. But how do they look at dynamical transitions between any given pair of IS? In their previous paper (Ref. 35), they talk about allowing their MD simulation to rapidly visit distinct IS, but there is no such discussion in this paper. Furthermore, since they investigate all pairs of IS, it is not at all clear that it is easy to get from one IS to its partner in the pair. The members of a pair of IS could be very different, making it hard to travel from one to the other. Some discussion needs to be added to explain how dynamical transitions are explored.

5. Following up on the previous point, what exactly is T_{ij} ? Is it a transition rate? In other words, what are the units of T_{ij} ?

We thank the Reviewer for pointing this out. The role of T_{ij} is actually very important to underline the effectiveness of our new ML approach. First, we now use labels α and β for IS instead of i and j which are instead used to refer to particles. We also clarified that α refers to the lowest energy minimum, and β the highest, which clarifies the distinction between $T_{\alpha\beta}$ and $T_{\beta\alpha}$.

We have clarified this in the manuscript:

"The transition matrix elements $T_{\alpha\beta}$ count how many times IS_α and IS_β are visited consecutively, i.e. IS_α is reached at time t , and IS_β at time $t + \tau_{\text{quench}}$. Overall, $T_{\alpha\beta}$ is a number that counts the number of transitions observed from IS_α to IS_β , with no physical dimensions. Since this quantity depends on the specific trajectories collected, it varies for different quenching rates and simulation times. Here, we demonstrate that one advantage of our ML approach over a brute-force approach, as employed in Ref. [35]], is to consider all IS pairs, not only those visited consecutively in the MD trajectory (characterized by $T_{\alpha\beta} > 0$), as potential TLS. This expands massively the pool of candidates, while ensuring that computational effort is targeted to IS pairs which are likely to be TLS."

So, $T_{\alpha\beta}$ is not an equilibrium transition rate, because there is an exponentially large number of states, so an equilibrium calculation is not possible. It follows that we measure $T_{\alpha\beta} = 0$ for most of the pairs, as seen in Fig. 6(a) in which the probability distribution peaks around $T_{\alpha\beta} + T_{\beta\alpha} = 0$.

Typos:

1. Page 1, top of second column: “has allowed to create computer glasses” would be better phrased “has allowed the creation of computer glasses”
2. Page 2, 1st column: “associated to defects” should be “associated with defects”
3. Page 4, 1st column: “each pairs of IS” should be “each pair of IS”

Thank you for your careful reading. We have corrected the typos.

4. Page 5, 1st column, last paragraph: In this paragraph the authors talk about discarding IS pairs when the number of transitions is less than 4. Then they say that this is based on the assumption that high transition rates between two IS is a good indicator of a DW. This is a bit confusing because one sentence talks about low transition rates and the next sentence talks about high transition rates. So it would be better perhaps to start by stating the assumption that high transition rates between two IS is a good indicator of a DW, and then say that therefore, they discard those pairs which have a low transition rate.

Following the suggestion of the Reviewer, we have rewritten the paragraph in the following way:

"For this reason, the authors introduced a filtering rule based on the assumption that high transition rates during the dynamic landscape exploration is a good indicator that the minimum transition path between two IS forms a double well. Therefore, Ref. [35]] discarded all pairs $\alpha\beta$ of IS such that the number of jumps $T_{\alpha\beta}$ (from low to high energy IS) and $T_{\beta\alpha}$ (high to low) during the MD exploration is smaller than four, i.e. $\min(T_{\alpha\beta}, T_{\beta\alpha}) < 4$."

5. Page 5, 2nd column, last paragraph: “focus the numerical effort to DW” should be “focus the numerical effort on DW”

6. Page 6, 2nd column, end of 1st paragraph: “with few glasses that contains” should read “with a few glasses contain”. Notice that there are 2 typos in this sentence.

Thank you. We have corrected the typos.

7. Page 7, Figure 5(a): $|\vec{r}_0 - \vec{r}_1|$, $|\vec{r}_0 - \vec{r}_2|$, and $\Delta\vec{r}_2$ are not defined in the text or in the caption. These should be defined in the caption.

8. Page 7, Figure 5(b): T_{ij} and T_{ji} are not defined until later in the text. Therefore, they should be defined in the caption.

We have added the following sentence to the caption of Fig.5:

"The features include the single particle displacements $\Delta\vec{r}_i$ (which decrease with increasing label i), their relative position compared to the most displacing particle $|\vec{r}_1 - \vec{r}_i|$ and the number of recorded transitions from the lowest to highest energy IS in the dynamic exploration $T_{\alpha\beta}$ (and $T_{\beta\alpha}$ from high to low-energy IS)."

9. Page 7, 1st column: The classical energy splitting ΔE should be defined in the text where it is used. The definition is buried in the Methods section at the end of the paper.

We now say that:

"We see that the most important feature is the classical energy splitting ΔE corresponding to the energy difference between two IS."

10. Pages 7-8: I am reiterating an earlier comment, but at the bottom of page 7 and the top of page 8, the authors talk about transition rates between IS and that many pairs of IS have no transition. But the authors never explain how they look for transitions.

Following the earlier comment, we have included a paragraph dedicated to the definition of $T_{\alpha\beta}$ and how we measure it. In the earlier comment we also discuss why many pairs have no transition.

11. Page 8, 1st column, line 2: “When ΔE is large we rarely” should have a comma so that it reads, “When ΔE is large, we rarely”

12. Page 8, 2nd column: end of 2nd paragraph: “batch of prediction” should be “batch of predictions”

13. Page 9, 1st column, 1st line under METHODS: “polydisperse mixture” should be “polydispersed mixture”

We have corrected the typos.

14. Page 9, 2nd column: The Berendsen thermostat should have a reference to the appropriate paper or book.

We now report the original reference.

15. Page 10, 1st column: What are q_2 and q_{12} ? These are never defined.

We now say more clearly that:

"This set [of features] consists of: potential energy, particle positions, averaged bond-orientational order parameters [57] determined via a Voronoi tessellation that we denote $\{q_2, q_3 \dots q_{12}\}$, and finally particle radii."

16. Page 10, 2nd column: Does QS stand for quantum splitting? The authors should define QS. Yes, it does. We now define it explicitly.

17. Page 12, Reference 12: “P. w. Anderson” should be “P. W. Anderson”

Corrected.

Reply to Reviewer #4

The paper is overall well written and the results are convincing. The ML used is simple but this is not the point of this paper; rather it provides a clear use case where ML can unlock science in a justified way, by allowing to pre-filter specific pairs of configurations which are relevant to describe glass's dynamics.

I would overall recommend acceptance of this paper provided the more detailed points below can be addressed. That being said, there may remain a question of impact of this paper (more precisely: what will be the impact of gaining access to more TLS pairs) which I am not fully confident on.

We are grateful to the Reviewer for their time and positive appreciation of our work. The Reviewer's insightful observations and helpful criticism allowed us to improve our manuscript.

Regarding impact, collecting an extensive number of TLS is a crucial step towards developing a microscopic theory to understand their properties. This is an important fundamental problem in condensed matter physics, as demonstrated by our work being highlighted in a recent commentary which appeared in the Journal Club for Condensed Matter Physics in January 2023. Gaining a better understanding of TLS, and how to tune their microscopic properties or density in real materials is also an important goal with multiple technological applications. For instance, TLS, present in amorphous layers forming q-bits, were shown to give rise to decoherence in quantum computers.

Below, we reply point-by-point to all comments and questions. Most importantly, for each point we have revised the manuscript accordingly, with changes highlighted in red to facilitate review.

More detailed comments:

1. Even though this is first and foremost a physics paper, I think that some details on the ML procedure and results could be explained a bit further, for instance (apologies if I missed some of those in the text): a. Giving some high level details of the type of models used in the main text rather than the name of the package used

We have added the following paragraph to give some details about the ML method used to classify DW:

"In particular, we get the best results using ensembles of gradient boosting models, which have proven to be the optimal choice in estimating barrier heights of chemical reactions computed with similar methods. The gradient boosting approach predicts the probability $p(y_i) = G(x_i)$ that a pair x_i is a DW, where $y_i = 1$ if the pair is a DW and 0 otherwise. It is based on a series of $m = 1, \dots, n_{WL}$ weak learners W_m that minimize the log-loss score

$$\frac{1}{n} \sum_i y_i \log [p(y_i)] - (1 - y_i) \log [1 - p(y_i)], \quad (1)$$

where n is the number of IS pairs. In this approach, each of the weak learners W_m attempts to improve over the result of its predecessor by predicting the residual $h_{m-1}(x_i) = y_i - W_{m-1}(x_i)$. The final prediction thus becomes

$$p(y_i) = \sum_{m=1}^{n_{WL}} W_m(x_i | W_{l < m}). \quad (2)$$

This approach usually outperforms random forests, where the prediction is just the average over the weak learners $p(y_i) = 1/n_{WL} \sum_{m=1}^{n_{WL}} W_m(x_i)$. We then build a stack of gradient boosting models such that the final prediction is given by $p(y_i) = 1/n_{GB} \sum_k^{n_{GB}} c_k G_k(x_i) / \sum_k c_k$, which is the weighted average over the n_{GB} models in the ensemble with learnable weights c_k ."

We then also detail the ML model used to predict the quantum splitting in the corresponding section, by adding:

"We thus follow the same strategy introduced for DW classification, by using model ensembling and gradient boosting. Compared to the DW predictor, each model G_k in the ensemble now performs a regression task by predicting $E_{qs,k} = G_k(x_i)$. We then construct a multi-layer stack (schematized in Fig. 2), where the prediction of the first stack $E_{qs}^{(0)} = 1/N \sum_k c_k G_k(x_i) / \sum_k c_k$ is concatenated to x_i and used as input for the following stack. At the same time, we also perform k -fold bagging, which consists in splitting the data in k subsets used to train k copies of each model with different data. This has shown to be particularly effective in improving the prediction for small datasets."

b. Reporting the accuracy/recall of the DW-classifier in the main text

We now report in the main text that: "Overall, our DW classifier turns to be very accurate and rapid, achieving $> 95\%$ accuracy after only 10 minutes of training."

c. Discussing the train/test split used for training and evaluating the model

We now report in sec. 2.C : "The train/test split is performed by randomly selecting 10% of the pairs to be used only for the evaluation."

e. It looks like the data is naturally described in log space – is this how the models were trained? If not, can this be an issue in relation with numerical accuracy? In particular can you discuss the larger errors near the left parts of the diagrams in Fig 1?

Yes, we train the model in log space. We believe that the error is larger near the left part because states with low quantum splitting are actually very rare (justifying our ML approach to find them!), thus they are underrepresented in the training set.

f. I was quite strongly surprised to see the MLP results in the Appendix, in particular the fact that the network does not match the mean of the prediction – could you discuss this further?

We have added the following paragraph to address this comment: "We believe that with sufficient training time and optimal hyperparameters, the MLP can perform better. However, our model ensembling approach is much faster and does not require any hyperparameter optimization, so we decided to not invest too much computational resources in the MLP. In particular, we believe that the model in Fig. S1(b) is overfocusing on the low quantum splitting pairs, that weight more in the loss function defined in eq.3."

2. On Figure 1 a. Could you discuss why the errors are higher at high temperature? Does it connect to Fig 4.c?

We think that the errors are higher at high temperature since there is more variance in the structure of the IS minima. This is partially visible in the histograms of Fig.4(c), where we show that at higher temperature there are glasses with few DW/TLS and glasses with many of them, while at low T the distribution has a significantly smaller variance.

b. Could you add a quantitative measure of correlation on the plots?

We now report the Pearson correlation coefficient in the corresponding panels of Fig.3. We added the sentence to the text "We see that the data concentrates around the diagonal, indicating good correlation between true and predicted values. The Pearson correlation reported in the figure provides a quantitative measure for the correlation."

c. Are the errors larger at lower energies and is that a potential concern?

The error at lower energies looks relatively larger only when we plot on a logarithmic scale. However this is not too much of a concern for the scope of the paper. Indeed, the ML model is mainly used to identify possible TLS, for which we then perform the exact calculations for the quantum splitting. We thus do not rely on the exact value predicted by the ML model. Instead what is important for our study is the confusion matrix, that we have color coded in Fig.4(a). We need to have as few false negatives as possible (blue quadrant), i.e. pairs of IS which are discarded by the ML filter, which are actually TLS. Our results confirm that the model is very performant in this respect.

3. On Figure 2: a. Could you explain what data is the confusion matrix based on and maybe giving number for it (otherwise it is very hard to read); also it is my understanding that the confusion matrix can only be based on pairs where $T_{ij}+T_{ji} \leq 4$, can we make an argument that it extrapolates to the general case?

First, we now use labels α and β for IS instead of i and j which are instead used to refer to particles. We also clarified that α refers to the lowest energy minimum, and β the highest, which clarifies the distinction between $T_{\alpha\beta}$ and $T_{\beta\alpha}$.

We now report the confusion matrix in Fig.4(a). The confusion matrix is already based on all the pairs because the ML algorithm can make its prediction even for pairs where $T_{\alpha\beta} + T_{\beta\alpha} \leq 4$, which turn out to be the majority of pairs (see Fig.6a).

b. I also did not understand what the topmost horizontal line was indicating.

The horizontal lines aimed to show two things: (i) how many true TLS can be identified by considering DW with a predicted QS below the line, (ii) how many NEB calculations are required to obtain the true QS.

Following the referee's advice, we decided to remove the topmost horizontal line for clarity. We now clarify the description of Fig. 4a in the main text: "The horizontal dashed lines highlight the percentage of true TLS detected by running the NEB algorithm for all points with a predicted quantum splitting below the line. Due to false negative, it is better to also consider transitions slightly above the TLS threshold. We see that all TLS are identified by considering only the pairs that are predicted to be within twice the quantum splitting threshold of TLS. All TLS thus are safely detected by running 2484 NEB calculations, out of 4147 DW in total."

3. On Figure 6.1, a. Would it make sense to report the cumulative fraction of split under a given $T_{ij}+T_{ji}$? That way one could check how to set the cutoff to gain a given fraction, yet not incur quadratic complexity.

We now report the cumulative fraction in Fig.6(a). However, we believe that setting a cutoff is not necessary since most of the pairs have small $T_{\alpha\beta} + T_{\beta\alpha}$, so there is no practical advantage in excluding the small fraction of data with large $T_{\alpha\beta} + T_{\beta\alpha}$.

b. I suppose the reason why there are splits for larger $T_{ij}+T_{ji}$ is because of the MD-induced walk in the energy landscape backtracking, is that correct? It may be worth saying in words (but maybe already said and I missed it).

This is correct. We in fact say that: "... TLS tend to originate from IS that are not too distant in order to have a reasonable tunneling probability. As such it is likely that those pairs of IS get explored one after the other during the exploration dynamics in step 2."

4. On the "quick and dirty" method a. Is there really a usecase for it given how quickly one can train your model? b. If there is, is it possible to come-up with a fixed criterion and evaluate its performance?

This is a fair point. We don't think there is a real usecase for the "quick and dirty" method in the TLS identification. However, we believe that the focus on the splitting may be useful to characterize similar questions that can not be directly addressed with our ML approach.

c. (related) Would it be possible on Table 2 to report the typical time taken by MD to generate a candidate IS pair? Unless it is \ll the NEB time, then the approach in this paper is less useful – is this correct?

We collected $\sim 10^7$ pairs in approximately $4.7 \cdot 10^7 [t_{MD}]$ at our lowest temperature. So on average, we generated a IS pair every $\sim 4.7 [t_{MD}]$. In a typical simulation we run $10^2 [t_{MD}]$ in 10^3 s, so $[t_{MD}] \sim 10$ s (wall clock time). This long computation time is due to the frequent energy minimization steps. We now report in Table 2 the average time to generate a candidate IS pair as "Collection of a IS pair: 50 s". Furthermore, we also realized that the previous value of NEB time was calculated using 40 CPUs, so we re-estimated the NEB time using the wall-clock time of a standard laptop. We do not believe that this time influences the effectiveness of the approach, because the NEBs remain a significant bottleneck, and contrarily to the collection of IS pairs, only a small fraction of NEBs are actually useful.

5. On the "iterative training", how is that different from what would generally fall under "active learning"?

We believe that 'iterative training' is not quite the same as 'active learning', since the iterations focus not on the samples with the highest variance, but instead on samples that are of physical interest. We now report that: "Compared to standard active learning methods, iterative training does not focus on new samples with the highest model uncertainty, but instead it iterates the predictions on the samples below the threshold of interest. "

REVIEWERS' COMMENTS

Reviewer #2 (Remarks to the Author):

I am satisfied with the authors response to my comments and questions. I think the paper should be published in Nature Communications once the few minor points below are clarified and some typos are corrected.

Comments and questions:

1. Page 3: What is the value of ϵ in Kelvin or eV? ϵ sets the energy scale of the pair interactions in the simulated model. The authors say they are using the numbers for argon, so what is ϵ for argon? They may have stated the value of ϵ , but I could not find it.
2. Page 6, caption for Figure 4: Should "0.0015" be "0.0015 ϵ "?
3. Page 6, second column: Should " $E_{qs} < 0.0015$ " be " $E_{qs} < 0.0015\epsilon$ "?

Typos:

1. Page 4, bottom of first column: "as input feature the number" should read "as input, the number"
 2. Page 10, bottom of first column: "consecutively, i.e. IS_{α} " should read "consecutively, i.e., IS_{α} ". Add a comma after "i.e."
 3. Page 11, second column: "turns to be very" should read "turns out to be very"
- Once these comments, questions and typos have been addressed, I recommend that the paper be published in Nature Communications.

Reviewed by Clare Yu.

Reviewer #4 (Remarks to the Author):

The authors have addressed the questions that I had, both in their response and in manuscript changes. I would now recommend the manuscript for publication.

Reply to Reviewer #2 (Remarks to the Author)

I am satisfied with the authors response to my comments and questions. I think the paper should be published in Nature Communications once the few minor points below are clarified and some typos are corrected. [...] Once these comments, questions and typos have been addressed, I recommend that the paper be published in Nature Communications.

We would like to express our sincere gratitude for the review of our manuscript. We greatly appreciate the Reviewer's questions and are pleased to hear that we have adequately addressed them, both in our response and through the changes made to the manuscript. We truly value the Reviewer's support and recommendation to publish the manuscript. We firmly believe that the Reviewer's feedback has significantly contributed to improving the quality of our work.

Comments and questions:

1. Page 3: What is the value of ϵ in Kelvin or eV? ϵ sets the energy scale of the pair interactions in the simulated model. The authors say they are using the numbers for argon, so what is ϵ for argon? They may have stated the value of ϵ , but I could not find it.

We have now added the following paragraph in the Methods section:

"We make two choices for physical units following past work, [35] one corresponds to Ar atoms [55]: $m = 6 \times 10^{-26}$ kg, $\epsilon/k_B = 478$ K, $\sigma = 3.41 \times 10^{-10}$ m and $\tau = 1.08 \times 10^{-12}$ s; the other is for a NiP alloy [56]: $m = 1.02 \times 10^{-25}$ kg (^{62}Ni isotope), $\epsilon/k_B = 4447$ K, $\sigma = 2.21 \times 10^{-10}$ m and $\tau = 2.86 \times 10^{-13}$ s."

2. Page 6, caption for Figure 4: Should "0.0015" be "0.0015 ϵ "?

Yes, we have now fixed the value.

3. Page 6, second column: Should "Eqs < 0.0015" be "Eqs < 0.0015 ϵ "?

Yes, we have added the missing unit.

Typos: 1. Page 4, bottom of first column: "as input feature the number" should read "as input, the number" 2. Page 10, bottom of first column: "consecutively, i.e. IS α " should read "consecutively, i.e., IS α ". Add a comma after "i.e.". 3. Page 11, second column: "turns to be very" should read "turns out to be very"

We thank the Reviewer for pointing this out. We have corrected all the mentioned typos.

Reply to Reviewer #4 (Remarks to the Author)

The authors have addressed the questions that I had, both in their response and in manuscript changes. I would now recommend the manuscript for publication.

We appreciate the Reviewer's questions and are pleased to hear that they have been adequately addressed. We are thankful for the recommendation to publish the manuscript, and we believe that the Reviewer's feedback has greatly contributed to improving the quality of their work.